

# Hydrological recovery in two large forested watersheds of Southeastern China: importance of watershed property in determining hydrological responses to reforestation

Wenfei Liu[1], Xiaohua Wei[2*], Qiang Li[2], Houbao Fan[1], Honglang Duan[1], Jianping Wu[1], Krysta Giles-Hansen[2],

Hao Zhang[1]

[1]Institute of Ecology and Environmental Science, Nanchang Institute of Technology, Nanchang, China
[2]Department of Earth and Environmental Sciences, University of British Columbia (Okanagan campus), 1177 Research Road, Kelowna, British Columbia, Canada, V1V 1V7

*Correspondence to*: Xiaohua Wei (adam.wei@ubc.ca)

**Abstract**. Understanding hydrological responses to reforestation is an important subject in watershed

management, particularly in large forested watersheds (>1000 km$^2$). In this study, we selected two large forested

watersheds (Pingjiang and Xiangshui) located in the upper reach of the Poyang Lake watershed, Southeastern

China (with an area of 3261.4 and 1458 km$^2$, respectively) to assess the effects of large-scale reforestation on

streamflow. Both watersheds have similar climate and experienced comparable and dramatic forest changes

during the past decades, but with contrasted watershed properties (e.g., the topography is much steeper in

Xiangshui than in Pingjiang), which provides us with a unique opportunity to compare the differences in

hydrological recovery in two contrasted watersheds. Streamflow at different percentiles (e.g., 5%, 10%, 50% and

95%) were compared using a combination of statistical analysis with year-wise method for each watershed. The

results showed that forest recovery had no significant effects on median flows ($Q_{50\%}$) in both watersheds.

However, reforestation significantly reduced high flows in Pingjiang, but had limited influence in Xiangshui.

Similarly, reforestation had significant and positive effects on low flows ($Q_{95\%}$) in Pingjiang, while it did not

significantly change low flows in Xiangshui. Thus, hydrological recovery is limited and slower in the steeper

Xiangshui watershed, highlighting that watershed property is also important for determining hydrological responses to reforestation. This finding has important implications for designing reforestation and watershed management strategies in the context of hydrological recovery.

Key words: large watershed, reforestation, streamflow changes, streamflow at different percentiles

## 1. Introduction

Water availability is of utmost importance for ecosystem functions, and economic and social development. In

forested watersheds, forests play an important role in hydrological processes and their associated ecological functions. Numerous studies have indicated that forest changes (e.g., reforestation or deforestation) can significantly affect hydrological processes (Jackson et al., 2005; Clinton, 2011; Ford et al., 2011; Iroumé and Palacios, 2013; Liu et al., 2015a). However, there are large variations in hydrological responses to forest changes, probably depending on climate and watershed characteristics. Understanding those variations can greatly improve

our understanding of the possible mechanisms responsible for hydrological responses and support our management decisions on water and watershed protections.

In large forested watersheds, various factors including climate, land cover or forest changes and watershed properties can influence streamflow (Anderson and Kneale, 1982). While previous research mainly focused on

how climate and forest cover change affect hydrology, limited research has been conducted to examine the role of watershed property in hydrological responses. However, watershed property can be an important factor in determining hydrological responses (Allan, 2004; Poff et al., 2006a; Poff et al., 2006b; Price et al., 2011; Troch et



al., 2013; Zhou et al., 2015). For example, Zhang et al. (2014) studied two neighboring watersheds (3420 km$^2$) and Willow (2860 km$^2$) in British Columbia, Canada, and found that their contrasted hydrological responses to forest harvesting are mainly related to the difference in their topography and landform complexities. Zhou et al (2015) also found that watershed characteristics such as watershed slope and size play an important role in

hydrological responses in their global analysis. Clearly, more case studies are needed to assess how watershed property affects hydrological responses in the context of the other key drivers (e.g., climate and forest changes).

Poyang Lake of Jiangxi Province directly flowing into Yangtze River is the largest freshwater lake (3500 km$^2$) in China. It is fed by five rivers including Gan, Xin, Xiu, Rao and Fu. Poyang Lake provides significant water

resources, wildlife habitats (especially for migratory birds) and economic values (Guo et al., 2008; Huang et al., 2012; Schmalz et al., 2014). However, Poyang Lake basin experienced severe forest disturbance from 1960s to 1980s. Such intense land use changes had resulted in severe environmental degradation. To restore degraded environment, several ecological restoration and protection programs (e.g., large-scale reforestation) have been implemented since 1980s (Wei et al., 2008). As a result, the forest coverage was increased significantly in the past

a few decades. Because Poyang lake basin plays a strategic significance in environmental protection and economic development in the province as well as in the lower reach of Yantze River basin, assessing the ecological effects of those large-scale stewardship programs would be crucial for determining the effectiveness of ecological recovery and for guiding future program design. To our knowledge, several studies had been conducted to assess how large-scale reforestation programs might affect soil erosion and forest carbon processes, but no

research has been conducted to assess hydrological recovery under those large-scale stewardship programs.

Two large neighboring watersheds including Pingjiang watershed (2689.20 km$^2$) and Xiangshui watershed (1758

km$^2$) with similar forest change levels but different watershed properties in the upper reach of the Poyang Lake watershed were chosen for the study. Hydrological variables such as streamflow at different percentiles (e.g., high flows and low flows) were examined for each watershed, and their differences were then compared. The objectives of this study were: (1) to assess how stream flows (high and low flows) responds to forest changes at

each watershed; (2) to compare their hydrological responses between two contrasted watersheds; and (3) to discuss implications for watershed management.

**2. Watershed descriptions and data**

**2.1. Watershed characteristics**

The Pingjiang and Xiangshui watersheds feed into Gan River, the largest tributary of the Poyang Lake watershed (Fig. 1). The drainage areas of the two watersheds are 2689 km$^2$ and 1758 km$^2$, respectively. The two watersheds are located in the hilly region of Jiangxi Province, China. The Xiangshui watershed is characterized with a steeper topography than the Pingjiang watershed with the former having 23.9% of the watershed area being higher slopes (from 30° to 50°) while the latter having only 4.6% for the same slope class (Table 1). Soils are mountain red soil

and yellow-red soil with sandy loam texture in both watersheds. The main characteristics of two watersheds are presented in Table 2.

**Figure 1.** The location of the Pingjiang and Xiangshui watersheds

**Table 1.** Averaged slopes in two studied watersheds (Pingjiang and Xiangshui)

**Table 2.** A summary of watershed characteristics for Pingjiang and Xiangshui watersheds

The two studied watersheds belong to subtropical monsoon zone with a similar precipitation regime. The average annual precipitations are 1575 mm and 1611 mm in Piangjiang and Xiangshui watersheds, respectively, of which

mostly fall from April to June (the wet season) and less from September to November (the dry season). The

average annual temperatures are 18.9 °C and 19.2 °C, respectively. The maximum temperature in summer and the

minimum temperature in winter are 37 °C and 0 °C, respectively (Fig. 2).

**Figure 2.** Average monthly precipitation, streamflow, maximum temperature and minimum temperature from

1957 to 2006 for the Pingjiang watershed (a) and the Xiangshui watershed (b)

The majority of annual peak flows corresponds to rainfall events in two watersheds. In Pingjiang watershed,

annual peak flows are between 368 and 1530 m$^3$/s, while between 291 and 1280 m$^3$/s in Xiangshui watershed.

Annual minimum flows range from 5.5 to 20.4 m$^3$/s in Pingjiang watershed, while are from 2.3 to 20.1 m$^3$/s in

Xiangshui watershed. Average annual mean flows are 848 and 858 m$^3$/s, respectively.

The major land cover types include forests, agriculture, grass, and urban and construction land. Subtropical

evergreen broad-leaved forest is the major climax vegetation type in the studied watersheds, including

*Castanopsis fabri*, *Castanopsis sclerophylla*, *Schima superba*, *Sassafras tzumu* and *Castanopsis fissa*. In contrast,

major plantation forests are *Pinus massoniana*, *Cunninghamia lanceolata*, *Camellia oleifera* Abel and *moso*

*bamboo*.

### 2.2. Data

Data on streamflow in two studied watersheds are available from 1957 to 2014. The hydrometric stations for data

collection are part of the Chinese National Hydrometric Network (Fig. 1). Climate data are also available for the

same length (1957-2014) for each watershed (5 climate stations for Pingjiang and 3 for Xiangshui), and include

the records of daily maximum, mean, and minimum temperatures and daily precipitation. The averaged

watershed-based precipitation estimates were derived by the Thiessen polygon method.

**3. Methods**

**3.1. Leaf area index (LAI) and forest coverage**

The Global Land Surface Satellite (GLASS) LAI data were used as the proxy of forest coverage in the studied

watersheds. The GLASS LAI product provides the global LAI at the spatial resolution of 0.05 degree and

temporal resolution of 8-day for the period of 1981 to 2014 (http://www.bnu-datacenter.com/). The GLSS

LAI data has been validated through the field measurements to ensure data quality for long term

studies in vegetation changes (Liang and Xiao, 2012; Xiao et al., 2014). The growing season LAI values were

based on the LAI values from April to October for each year. The watershed-based LAI values were derived by

averaging the LAI data for the pixels where more than 50% of their pixel areas falls inside the watershed

boundaries.

Forest change is the main type of land use changes in our studied watersheds. Because the complete records of

annual deforestation and reforestation areas are unavailable, forest coverage and LAI data were used to indicate

historic forest changes during the study period (1957-2006). As shown in Figure 2, forest cover was greatly

reduced in 1965-1984 due to large-scale forest disturbance (e.g. deforestation). Since then, forest cover was

significantly increased from about 30% in the 1980s to 70% in 2006 in both watersheds due to implementation of

the reforestation projects (1990-2006) (Figure 3). Thus, the entire study period was divided into the forest

disturbance period (1957-1985) and the forest recovery period (1990-2006).

**Figure 3.** Forest coverage (%) from 1950 to 2006 in the Pingjiang and Xiangshui watersheds (a) and (b) Leaf

Area Index (LAI) for the Pingjiang and Xiangshui watersheds

### 3.2. Median, high and low flows

In this study, FDCs (flow-duration curves) were applied to define high, median and low flows. FDCs represent the

percent of time streamflow for any given value exceeded or equaled in a period of record (Vogel and Fennessey,

10    1994). In this study, median flows are defined as the flows that exceed or are equal to $Q_{50\%}$. High flows are

defined as the flows that exceed or are equal to $Q_{5\%}$ and $Q_{10\%}$ ($Q_{5\%}$: flows exceeded at 5% of the time in a given

year and $Q_{10\%}$: flows exceeded at 10% of the time in a given year), while low flows are defined as the flows that

are equal to or less than $Q_{95\%}$ ($Q_{95\%}$: flows exceeded at 95% of the time in a given year) (Zhang et al., 2014b; Liu

et al., 2015b).

In order to assess the impacts of forest changes on high, median and low flows, the effect of climate variability

must be eliminated. For a single watershed, pair-wise comparisons can be used to address this issue (Levy, 1975;

Broomell et al., 2011; Zhang et al., 2014b; Liu et al., 2015b; Eastwood et al., 2016). Because high flows are

mainly caused by some rainfall events, we can find some similar and comparable rainfall events between the

20    reforestation and deforestation periods with similar $R_{5\%}$ and $R_{10\%}$, respectively ($R_{5\%}$: rainfall exceeded at 5% of

the time in a given year and R10%: rainfall exceeded at 10% of the time in a given year). However, low and

median flows are significantly correlated with annual rainfall, annual maximum temperature and annual mean





temperature (Tables 3 and 4). Therefore, paired years between the reforestation and deforestation periods were

selected for low flow analysis (Tables S1 and S2). More details about this method can be found in Zhang and Wei

(2014a) and Liu et al. (2015b).

**Table 3.** Correlation analyses between low flows and climatic variables in the Pingjiang and Xiangshui

watersheds

**Table 4.** Canonical correlation analyses between climate variables and hydrological variables (low and median

flows) in the Pingjiang and Xiangshui watersheds

**Table S1.** Selected pairs for high flows (5% and 10%) in the Pingjiang and Xiangshui watersheds

**Table S2.** Selected pairs for median and low flows in the Pingjiang and Xiangshui watersheds

### 3.3. Estimation of recession constants

Recession constant is a useful indicator reflecting the characteristics of the study basin (Barnes and Bertram, 1939;

Ge et al., 2014). For a watershed, the difference in recession constants of streamflow with similar climate

conditions between different periods can be ascribed to the effect of land cover change, while the difference in

recession constants of streamflow between two studied watersheds under similar climate conditions can be

ascribed to the effect of different water properties on streamflow.

In this paper, the classical recession curve based on Genetic Algorithm (GA) was adopted to study and analyze the

daily runoff (Equations 1 and 2).

$$Q_t = Q_0\, e^{-\beta t} \tag{1}$$





$$\beta = (\ln Q_0 - \ln Q_t)/t \tag{2}$$

Here, $Q_0$ is the initial discharge (t = 0), $Q_t$ is the discharge at a later time $t$ (usually in days), and $\beta$ is the recession constant.

5 The paired-wise approach was also used to assess the effects of forest changes on recession constants. Because high flows are mainly caused by rainfall events (e.g., storm events) in the study area, we can select similar and comparable rainfall events between the forest reforestation and the disturbance periods (Table S3).

**Table S3.** Selected pairs for recession constant in the Pingjiang and Xiangshui watersheds

## 4. Results

### 4.1. High flows response to forest changes

As shown in Figure 4a, the average magnitude of high flows ($Q_{5\%}$) in the reforestation period (327.7 m³/s) was
15 significantly lower ($p<0.01$) than that in the deforestation period (534.9 m³/s) in the Pingjiang watershed. Similarly, the average magnitude of high flows ($Q_{10\%}$) in the reforestation period (164.4 m³/s) was also significantly lower ($p<0.01$) than that in the deforestation period (198.7 m³/s) in the Pingjiang watershed (Fig. 4b).

20 For the Xiangshui watershed, the average magnitude of high flows ($Q_{5\%}$) in the reforestation period (233.0 m³/s) was lower than that in the deforestation period (251.4 m³/s) (Fig. 4c), but their difference was not statistically significant (p=0.46). The average magnitude of high flows ($Q_{10\%}$) in the reforestation period (118.0 m³/s) was

significantly lower ($p<0.05$) than that in the deforestation period (127.9 m³/s) (Fig. 4d). Thus, reforestation

significantly decreased high flows in the Pingjiang watershed, while such an effect is relatively limited in the

Xiangshui watershed.

5 **Figure 4.** High flows and median flows for the selected pairs in the reforestation and deforestation periods: (a)

high flows (5%) for the Pingjiang watershed; (b) high flows (5%) for the Xiangshui watershed; (c) high flows

(10%) for the Pingjiang watershed; (d) high flows (10%) for the Xiangshui watershed; (e) Median flows (50%) for

the Pingjiang watershed; and (f) Median flows (50%) for the Xiangshui watershed

10 **4.2. Median flows response to forest changes**

As shown in Figures 4e and 4f, the averaged magnitudes of median flows in the reforestation period (43.1 and

41.5 m³/s, respectively) were marginally higher ($p=0.21$ and 0.27, respectively) than those in the deforestation

period (40.3 and 38.4 m³/s) in Pingjiang and Xiangshui watersheds, respectively, indicating that reforestation had

15 no significant effects on median flows ($Q_{50\%}$) in both watersheds.

**4.3. Low flows response to forest changes**

As shown in Figure 5a, the average magnitude of low flows in the reforestation period (12.3 m³/s) was

20 significantly higher ($p<0.01$) than that in the deforestation period (8.7 m³/s) in the Pingjiang watershed. In

contrast, the average magnitude of low flows in the deforestation period did not significantly differ from that in

the reforestation period (Figure 5b) in the Xiangshui watershed. Thus, reforestation significantly increased low

flows in the Pingxiang watershed but not in the Xiangshui watershed.

**Figure 5.** Low flows and recession constants of streamflow for the selected pairs in the reforestation and deforestation periods: (a) low flows for the Pingjiang watershed; (b) low flows for the Xiangshui watershed; (c) recession constants for the Pingjiang watershed; and (d) recession constants for the Xiangshui watershed

**4.4. Responses of recession constants to forest changes**

As shown in Figures 5c and 5d, the averaged recession constant of streamflow in the reforestation period was significantly lower (p=0.049) than that in the deforestation period in the Pingjiang watershed, while the difference was not significant (p=0.52) in the Xiangshui watershed, suggesting that hydrological responses to reforestation is more sensitive in the Pingjiang watershed than in the Xiangshui watershed.

**5. Discussion**

Although the effects of reforestation on peak flows are still controversial (Gafur et al., 2003; Nadal-Romero et al., 2016; Liu et al., 2015a), a general conclusion is that increased forest coverage through reforestation can reduce high flows (Llorens et al., 1997; Gebrehiwot et al., 2010; Nadal-Romero et al., 2016). Our study found that reforestation can significantly decrease high flows, which can thereby reduce flood risks. Thus, our results are consistent with the general conclusion conducted in other regions (e.g., Gafur et al., 2003; Bahremand et al., 2007; Tran et al., 2010). Our results are also supported by another study in a neighboring watershed (Meijiang) of the same region (Liu et al., 2015b) where the historic forest change is similar to those in our study. The common

reason for reducing high flows after reforestation is that reforestation increases forest coverage and slowly improves soil conditions, and consequently enhance soil infiltration capacity and reduce high flows.

Our study showed that reforestation significantly increased low flows in the Pingjiang watershed. Although not statistically significant, the low flows after reforestation in the Xiangshui watershed was also improved (Figure 5b). Thus, reforestation had a positive role in low flows in the study watersheds. Our results are consistent with various reforestation studies, particularly in higher humidity environment (Buttle, 2011; Yao et al., 2012; Liu et al., 2015b). For example, Zhou et al. (2010, WRR) studied the effects of large-scale reforestation on hydrology in the whole Guangdong province, and found that increasing of 30% forest cover played a positive role in redistributing water from the wet season to the dry season and, consequently, in increasing water yield in the dry season. The main reason for enhancing low flows from reforestation is that reforestation improves vegetation and soil conditions, and consequently improves soil infiltration and groundwater recharging, which have positive effects on low flows.

The responses of low flows to reforestation are inconsistent across different climate regimes. Lu et al. (2016) firstly estimated effects of reforestation on groundwater resource using seven evapotranspiration models and suggested that China's unprecedented reforestation program would result in greatly decreased depth of groundwater in the arid and semiarid areas of northern China. A similar study conducted in the Loess Plateau of China also found that a statistically significant ($p < 0.1$) reduction of 0.03 mm of groundwater per year from 1955 to 2010 due to implementation of large-scale reforestation projects (Gao et al., 2015). The results from a paired watershed experiment in South African showed that low flows were reduced by half due to reforestation (Smith and Scott, 1992). A study analyzing the responses of streamflow to forest plantation expansion in six large river



watersheds (from 94 to 1545 km$^2$) of Central-Southern Chile indicated that reforestation had less effects on low

water flows ($Q_{80\%}$ to $Q_{90\%}$) in relatively drier soils (Iroumé and Palacios, 2013). However, in humid region,

increases in vegetation cover often lead to greater infiltration of rainfall into the soil, and as a result, increase

water storages and low flows (Zhou et al., 2010). More case studies are needed before a general conclusion

between reforestation and low flows can be developed.

Although reforestation generally played a positive role in streamflow in our study area, there are large differences

in the hydrological responses between two study watersheds. As shown above, there are more significant effects

on both high and low flows in Pingjiang watershed than in Xiangshui watershed. Since both watersheds have the

similar historic forest change and climate, we believe that the difference in their responses of high and low flows

was mainly due to the difference in their watershed properties. A close examination on their watershed properties

shows that their main differences in watershed property are on watershed slopes and sizes. Many studies show that

watershed size can be an important factor affecting hydrological responses to land cover changes (Buttle and

Metcalfe, 2000; Blöschl et al., 2007; Zhang and Wei, 2014a; Zhou et al., 2015). A smaller sized watershed often

has less buffering capacity as it may contain fewer heterogeneous landscape components (e.g., wetlands, lakes)

and complexities, and as a result, is more sensitive to land cover changes. In our study watersheds, Xiangshui

watershed is much smaller than Pingjiang watershed so a quicker hydrological response should be expected in

Xiangshui watershed. The limited and slower hydrological response in Xiangshui watershed after reforestation as

compared with Pingjiang watershed suggests that the factor other than watershed size came into play. Thus, we

reasonably judge that the difference in watershed slope between two watersheds is the major factor determining

the variations of their hydrological responses. The Xiangshui watershed has a much larger area percentage (23.9%)

with the slope class (30%-50%) as compared to that (4.6%) in the Pingjiang watershed (Table 1). In Southern

China where a monsoon climate is dominant, a steeper watershed often has more severe soil erosion if

deforestation occurs, and consequently it would take much longer time to recover through reforestation process

once severe soil erosion occurred (Chen et al., 2002; Zheng et al., 2015).

The importance of watershed property in hydrological responses to land cover or forest changes is gradually

recognized in scientific communities. This is particularly relevant for larger watersheds (e.g., >1000km2) where

there are more landforms (e.g., wetlands, lakes), more land cover types and thus more interactions and

complexities of various watershed properties. Several studies on forest changes and hydrology in large forested

watersheds in British Columbia, Canada conclude that the effects of forest changes on water are likely watershed

specific (Lin and Wei, 2008; Zhang and Wei, 2014b) which clearly demonstrates the importance of watershed

property in determining the relationship between forest changes and water. However, assessing how watershed

property affects hydrological responses among other key drivers such as forest change and climate is a

challenging subject. Some studies have applied integrative indicators such as topographic index (Woods et

al., 1997; Hjerdt et al., 2004; Liu et al., 2012) or flow paths and transit time (McGuire and McDonnell, 2006;

Soulsby et al., 2009) to assess watershed behaviors or functions while other studies used a landscape approach

(Poff et al., 2006a; Poff et al., 2006b; Price et al., 2011) Nevertheless, more case studies are needed in this

direction.

Our results from this study have important management implications. The Pingjiang and Xiangshui watersheds

are very important headwater systems to Poyang Lake, the largest freshwater lake in China, where is crucial to

sustain aquatic ecological functions (Guo et al., 2008). Many studies had demonstrated alteration of flow regimes

(especially for low and high flows) may be one of the most serious and ongoing threats to the integrity of river

ecosystems (Ward et al., 1999; Bunn and Arthington, 2002; Poff and Zimmerman, 2010; Liu et al., 2015b). Therefore, it is highly important to manage flow regimes for sustainable watershed ecosystems in Poyang Lake Bain.

5    Our results demonstrate a positive effect of reforestation on high and low flows in both Pingjiang and Xiangshui watersheds. This confirms that our reforestation programs implemented over the last decades provide important benefits to restoration of watershed functions in terms of hydrology. More importantly, our study found that hydrological recovery of a steeper watershed likely takes much longer time once it is deforested or damaged, suggesting that we must take extra care when we design management strategies in more sensitive watersheds.

## 6. Conclusion

We found that reforestation decreased high flows, but increased low flows in our studied watersheds, which is beneficial to maintenance of aquatic functions and water supply. We also found that there are large variations in

15    hydrological responses to similar reforestation levels likely due to the difference in watershed property (e.g., watershed slope). Thus, we conclude that hydrological recovery through reforestation is largely dependant on watershed property when forest change and climate are similar and comparable.

## Acknowledgments

20    Funding was provided by Jiangxi Education Department (No. GJJ151141), the National Science Foundation of China (No. 31170665), Jiangxi Education Department (KJLD12097 and KJLD14095), Gan-Po 555 Talent Project, and Funding of Jiangxi Province, Scientific Funding by Jiangxi Province (No. 20142BAB214006 and





20161BBH80049).

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





**Table captions**

Table 1 Averaged slopes in two studied watersheds (Pingjiang and Xiangshui)

| Watershed | Percentage of watershed area (%) | | | | | |
|---|---|---|---|---|---|---|
| | Slop 30°-50° | 20°-25° | 15°-20° | 12°-15° | 2°-5° | < 2° |
| Pingjiang | 4.60 | 52.82 | 2.40 | 29.44 | 6.63 | 4.11 |
| Xiangshui | 23.85 | 26.99 | 9.20 | 33.05 | 6.17 | 6.91 |





Table 2 A summary of watershed characteristics for the Pingjiang and Xiangshui watersheds

| Metrics | Pingjiang | Xiangshui |
| --- | --- | --- |
| Drainage area (km$^2$) | 2689.20 | 1758 |
| Average elevation (m) | 298 | 429 |
| Soil type | Mountain red soil and Yellow-red soil | Mountain red soil and Yellow-red soil |
| Annual mean precipitation (mm) | 1575 | 1611 |
| Annual mean Temperature (°C) | 18.9 | 19.2 |
| Annual mean ET(mm) | 879.2 | 936.8 |
| Annual mean flow(mm) | 848 | 858 |
| Runoff Coefficient | 0.54 | 0.53 |
| Maximum flow(m$^3$/s) | 1530 | 1280 |
| Minimum flow(m$^3$/s) | 5.5 | 2.3 |
| BioGeoClimatic zone | Subtropic monsoon | Subtropic monsoon |
| Forest type | Subtriopical everygreen boradleaf forest and conifer forest | Subtriopical everygreen boradleaf forest and conifer forest |
| Dominant Disturbance Type | Logging | Logging |
| Hydrometric station | Hanlinqiao | Mazhou |





Table 3 Correlation analyses between low flows and climatic variables in the Pingjiang and Xiangshui watersheds

| Watersheds | Precipitation | | Tmax | | Tmin | | Tave | | Wind speed | |
|---|---|---|---|---|---|---|---|---|---|---|
| | Manna-Kendall | Spearman | Manna-Kendall | Spearman | Manna-Kendall | Spearman | Manna-Kendall | Spearman | Manna-Kendall | Spearman |
| Pingjiang | 0.48** | 0.68** | -0.34** | -0.48** | 0.14 | 0.18 | -0.21* | -0.32* | -0.12 | -0.21 |
| Xiangshui | 0.57** | 0.39** | -0.44** | -0.61** | 0.04 | 0.04 | -0.23* | -0.05 | -0.07 | -0.11 |

Tave, Tmax and Tmin refer to annual mean, maximum and minimum temperatures, respectively.

**Statistical difference at $p<0.01$.

*Statistical difference at $p<0.05$.





Table 4 Canonical correlation analyses between hydrological variables (median and low flows) and climatic

5      variables in the Pingjiang and Xiangshui watersheds

| Watersheds | Canonical correlation analysis | Canonical R | Significant |
|---|---|---|---|
| Pingjiang | | 0.88 | $p<0.01$ |
| | Precipitation, Tave and Tmax | | |
| Xiangshui | | 0.89 | $p<0.01$ |





Figure 1. The location of the Pingjiang and Xiangshui watersheds



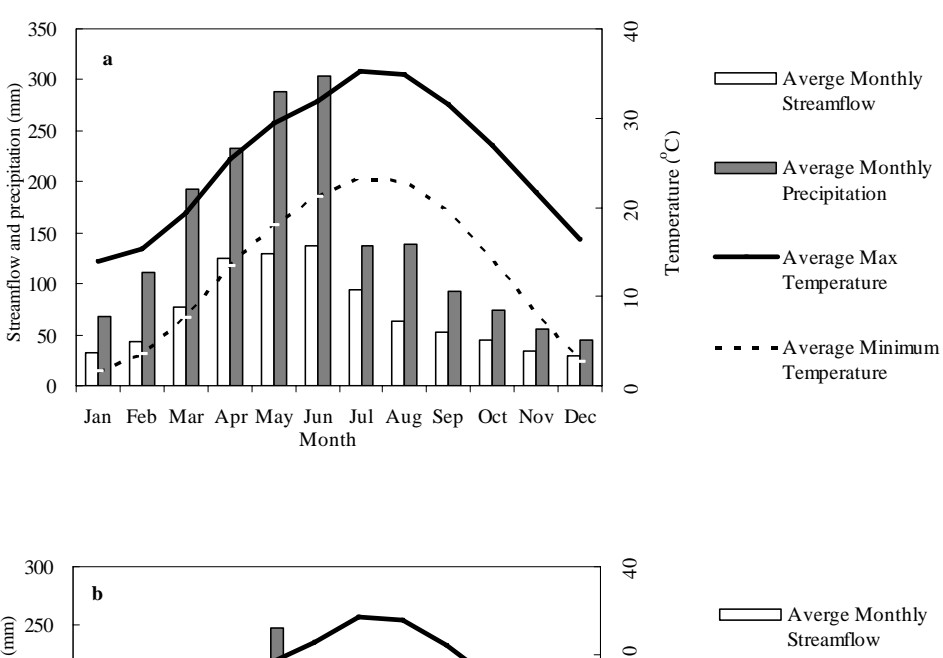

**Figure 2** Average monthly streamflow, precipitation, minimum temperature and maximum temperature from 1957

15    to 2006 for the Pingjiang watershed (a) and the Xiangshui watershed (b)





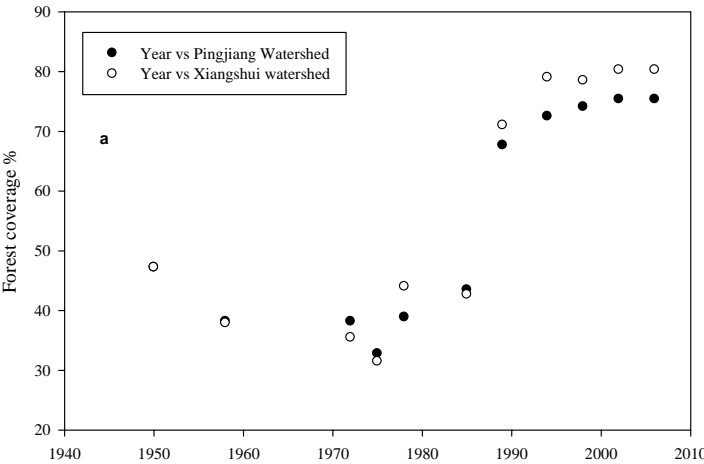

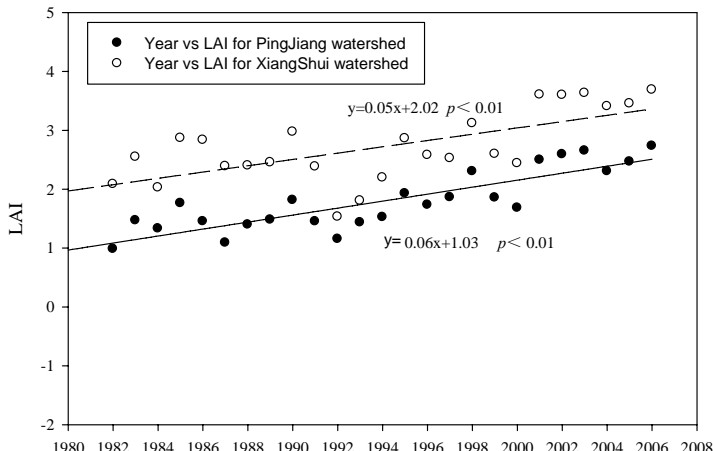

**Figure 3** Forest cover (%) (a) and Leaf Area Index (LAI) (b) from 1982 to 2006 in the Pingjiang and Xiangshui
10    watersheda





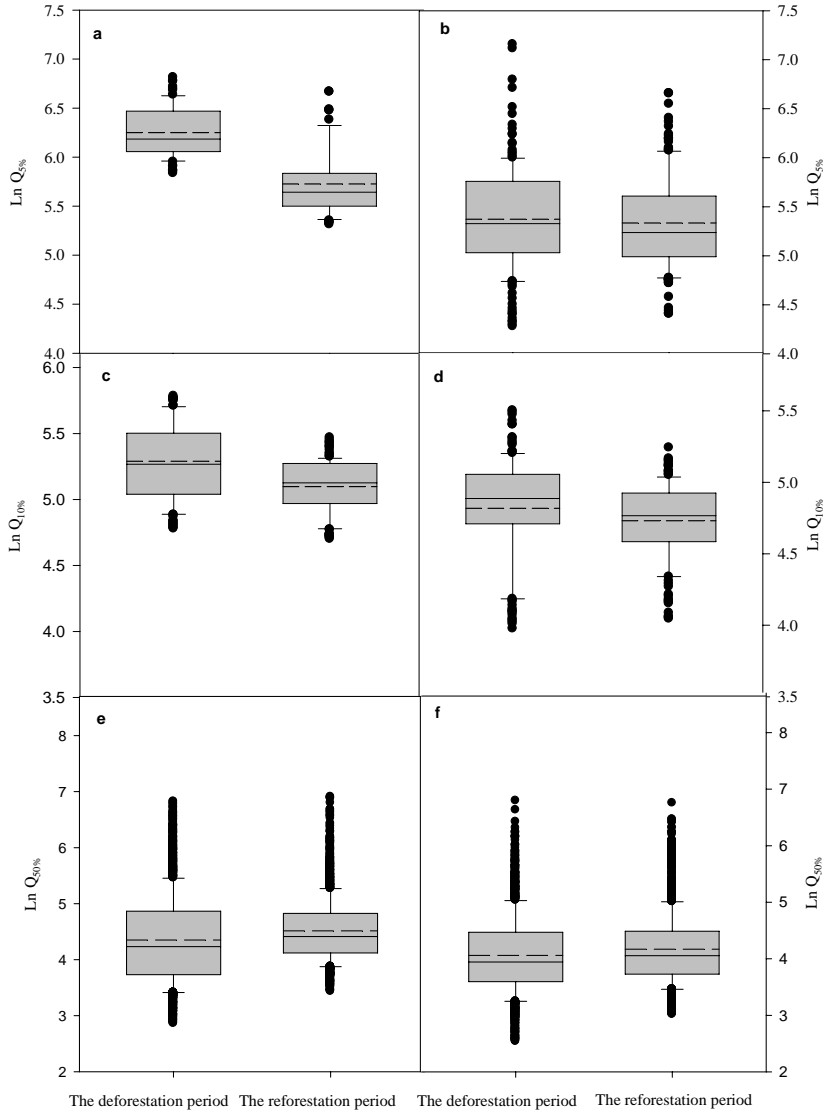

**Figure 4** High flows and median flows for the selected pairs in the deforestation and reforestation periods: (a)

high flows ($Q_{5\%}$) for the Pingjiang watershed; (b) high flows ($Q_{5\%}$) for the Xiangshui watershed; (c) high flows

($Q_{10\%}$) for the Pingjiang watershed; (d) high flows ($Q_{10\%}$) for the Xiangshui watershed; (e) Median flows ($Q_{50\%}$)

for the Pingjiang watershed; and (f) Median flows ($Q_{50\%}$) for the Xiangshui watershed





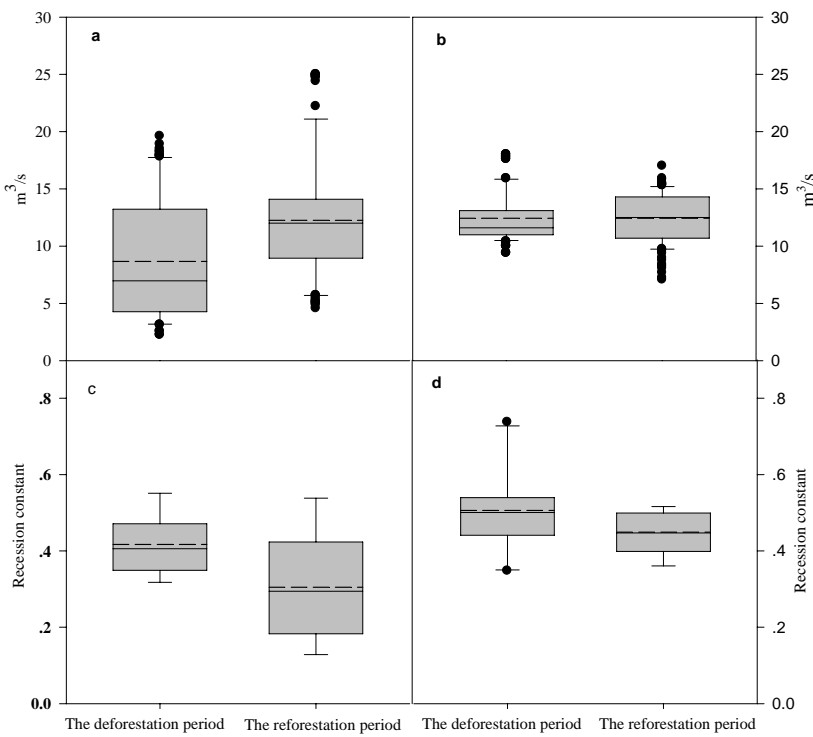

**Figure 5.** Low flows and recession constants of streamflow for the selected pairs in the deforestation and

reforestation periods: (a) low flows for the Pingjiang watershed; (b) low flows for the Xiangshui watershed; (c)

recession constants for the Pingjiang watershed; and (d) recession constants for the Xiangshui watershed

