# Peer review of "Hydrological recovery in two large forested watersheds of Southeastern China: importance of watershed property in determining hydrological responses to reforestation"

_Hydrology and Earth System Sciences, 2016_

## Referee Comment (RC1) · Anonymous Referee #1 · 9 Jul 2016

This paper detected and compared the different responses of high- median- and low-flow of 2 similar watersheds to forestation using long-term data. Based on this analysis, the authors stated that besides climate and vegetation, the watershed properties of topography (such as the slope gradients) also play an important role in the runoff response to forestation. This finding is correct and important for promoting the accurate evaluation of forest impact on watershed runoff and for guiding the watershed management.

I have some scientific/technical questions/suggestions as below for the authors: 1) In

the abstract, it is better to give the period (years to years) of data used. 2) If there is data about the difference in soil thickness (also soil physical properties) of the 2 watersheds, they can be listed in the watershed description and used in the discussion. Or if the author can find some literature about the relation between slope gradient and soil thickness (water-holding capacity), they can be used in the discussion to support your main conclusion. 3) P2L9: Since the study area is in humid region, I suggest at the beginning of this paper to say that "the water quantity and time distribution is of utmost importance ....", instead of "Water availability"? 4) It is necessary to say if the watersheds were obviously affected by reservoir construction? or have the same effect? 5) P5L8-11: Since the big difference in watershed area, it is better to give the flow discharges per unit watershed area for an easier comparison. 6) P9L20-22: If you look at the difference in the extreme high flows between the deforestation and reforestation periods, you can see the forest impact. You may add some words about this in your research result and conclusion. 7)Fig. 3: I doubt the too low LAI in watershed Xiangshui, just between 1-2.5? It may also low in Pingjiang? Please check. 8) P7L20-21: Delete the words in parentheses which is a repeat of former explanation (L11-12). 9) P8L1-2: Paired years were selected not only for low flow, but also for high and median flow. Correct the text. 10) P9L7: Delete "forest" before "reforestation". 11) P9L18: Replace "Fig. 4b" with "Fig. 4c". 12) P9L21: 4c replaced by 4b. 13) P14L16: Add "." before "nevertheless". 14) Table 1: Please check the slope range classification. Why not continuously?

---

## Referee Comment (RC2) · Anonymous Referee #2 · 15 Aug 2016

Dear Editor,

Thank you for the opportunity to review this interesting and informative manuscript. I believe your readership will find the paper valuable as it presents novel findings for a topic area often overlooked in the literature. Consequently, I recommend the manuscript be accepted for publication following some technical corrections. For ease of presentation I present them below:

- page 1 line 18 different vs contrasted - page 1 line 18 - state the different watershed properties - is it more than slope? - page 1 line 22 - explain year-wise method or be

more general in the description - page 1 line 14-15 - explain concept of hydrologic recovery associated with tree growth. - page 3 line 7 - properties vs. property - page 3 line 12 - define global analysis - page 3 line 14 - is it climate or weather - there are scale differences. - page 3 line 15 "which directly flows into the Yangtze...." - page 4 line 5 - "reach of the Yangtze" - page 4 line 5 - is or would be? - page 4 line 7 - have been vs. are - page 4 line 16 respond vs. responds - page 4 line 16 different vs. contrasted - page 5 line 14 "... watersheds are within the subtropical monsoon zone and have a similar precipitation regime." - page 5 line 16 - most vs mostly and also please provide rough proportions for seasonal precipitation - page 6 line 6 correspond vs (s) - page 6 line 7 "Pingjiang watershed and are 2.3 to 20.1 m3/s in the Xiangshui watershed. - page 6 line 16 Stream flow data area available from 1957 to 2014 for both watersheds. -page 7 line 12 onwards - font - page 8 line 13 - median is not capitalized page 12 line 4 - p=0.21 and 0.27 ? How are these p-values associated with a marginally higher median flow? - page 12 section 4.3 - would benefit from details on regrowth metrics such as canopy closure or height and a comparison between basins. - page 13 line 4 0.049 - page 13 line 18 - s after enhance and reduce - page 14 - line 2 - how were low-flows improved? - page 14 line 17 - South Africa - page 15, line 3 - humid regions - on another note curious that increase in vegetation leads to greater net precipitation - page 15 line 8 - between the two studied.... - page 16 line 10 - careful with this statement because watershed characteristics/properties such as slope have long been recognized as having a significant influence on runoff.

---

## Author Comment (AC1) · 16 Sep 2016

**Responses to the reviewers' comments**

Re: Hydrological recovery in two large forested watersheds of Southeastern China: importance of watershed property in determining hydrological responses to reforestation

**Reviewer: 1**

1. In the abstract, it is better to give the period (years to years) of data used.

**Response:** Done as suggested.

2. If there is data about the difference in soil thickness (also soil physical properties) of the 2 watersheds, they can be listed in the watershed description and used in the discussion. Or if the author can find some literature about the relation between slope gradient and soil thickness (water-holding capacity), they can be used in the discussion to support your main conclusion.

**Response:** We have carefully considered the reviewer's suggestion. Unfortunately, soil thickness usually exhibits a large spatial variability especially for large watersheds, and this data are unavailable for our studied watersheds. As we all know, when soil types are similar in the two watersheds, water-holding capacity in a flatter terrain is greater than that in a steeper one because the former has slower surface runoff. Although data on soil thickness is unavailable, the difference in slope between the two studied watersheds can also indicate variations of water-holding capacity.

3. P2L9: Since the study area is in humid region, I suggest at the beginning of this paper to say that "the water quantity and time distribution is of utmost importance ....", instead of "Water availability"?

**Response:** Done as suggested.

4. It is necessary to say if the watersheds were obviously affected by reservoir construction? or have the same effect?

**Response:** As far as we know, there are no large reservoirs were constructed in the upper reach of studied watersheds. Thus, the effects of reservoir construction on flow regimes can be ignored.

5.  P5L8-11: Since the big difference in watershed area, it is better to give the flow discharges per unit watershed area for an easier comparison.

**Response:** We agree with the reviewer's suggestion. We have calculated those data per 1000 km$^2$ for easier comparisons. Our manuscript has been upgraded accordingly.

6.  P9L20-22: If you look at the difference in the extreme high flows between the deforestation and reforestation periods, you can see the forest impact. You may add some words about this in your research result and conclusion.

**Response:** We agree with the reviewer's suggestion. We have added some descriptions which can be found in P15.line5-9.

7. Fig. 3: I doubt the too low LAI in watershed Xiangshui, just between 1-2.5? It may also low in Pingjiang? Please check.

**Response:** Thank the reviewer for pointing this out. The data on LAI in the two studied watershed were obtained from the global LAI at the spatial resolution of 0.05 degree and temporal resolution of 8-day for the period of 1981 to 2014 (http://www.bnu-datacenter.com/). We agree that our LAI may be underestimated because those estimates are from remote-sensing based methods. However, because the LAI data were mainly used to show forest change history and were not involved in any calculations, they would not affect our comparisons.

8. P7L20-21: Delete the words in parentheses which is a repeat of former explanation (L11-12).

**Response:** Descriptions in Line 11-12 described the definition of high flows, while the statements in Line 20-21 were used for locating some similar and comparable rainfall events between the reforestation and deforestation periods to eliminate the effects of climate variability on high flows. Thus, they are not a repeat, and they served difference purposes.

9. P8L1-2: Paired years were selected not only for low flow, but also for high and median flow. Correct the text.

**Response:** Thank the reviewer for pointing this out. Paired years were selected for low and median flows, while some similar and comparable rainfall events were selected to make pairs for high flows.

10. P9L7: Delete "forest" before "reforestation".

**Response:** Done as suggested.

11. P9L18: Replace "Fig. 4b" with "Fig. 4c". P9L21: 4c replaced by 4b. P14L16: Add "."

before "nevertheless".

**Response:** Thank the reviewer for careful reading. We have corrected these mistakes.

12. Table 1: Please check the slope range classification. Why not continuously?

**Response:** Thank the reviewer for pointing this out. After checking the slop range classification, we have updated Table 1 accordingly.

---

## Author Comment (AC2) · 27 Sep 2016

**Reviewer 2**

1. page 1 line 19 different vs contrasted

**Response:** Done as suggested (Page 1.line 19)

2. page 1 line 19 - state the different watershed properties - is it more than slope?

**Response:** In this paper, watershed area is regarded as another factor affecting hydrological responses to forest changes.

3. page 1 line 22 - explain year-wise method or be more general in the description

   **Response:** This approach matches years in the reference period with their comparable years in the disturbed period according to their similarities in climate conditions in each watershed. Thus, the hydrological difference between those selected years of two periods can be regarded as the effects of forest disturbance. **Please see our** explanations on Page7. line 16-22 and Page 8. line 1-3.

4. page 1 line 21 - explain concept of hydrologic recovery associated with tree growth

**Response:** Hydrological recovery is defined as the process that hydrological functions in a disturbed watershed are restored toward pre-disturbed condition by forest regeneration. Forests affect water through interception, evaporation, and transpiration etc. After forest disturbance (e.g., forest harvesting), hydrological processes are affected, and then will be gradually recovered with forest re-growth.

5. page 3 line 6 - properties vs. property

**Response:** Done as suggested (Page 3.line 6)

6. page 3 line 12 - define global analysis

**Response:** Thank the reviewer for this concern. We have added descriptions in Page 3. line 5-6.

7. page 3 line 14 - is it climate or weather-there are scale differences

**Response:** It is climate. In forested watersheds, forest change and climatic variability are commonly regarded as two major drivers for influencing hydrological variations.

8. page 3 line 8 "which directly flows into the Yangtze...."

**Response:** Done as suggested (Page 3.line 9)

9. Page 3 line 12 – remove "had" –

**Response:** Done as suggested (Page 3.line 13)

10.  Page 3 Line 14 – consider "Forest cover has increased…"

**Response:** Done as suggested (Page 3.line 15)

11.page 4 line 5 consider different vs. contrasted

**Response:** Done as suggested (Page 4.line 6)

12. page 4 line 21   consider"... watersheds are within the subtropical monsoon zone and have a similar precipitation regime."

**Response:** Done as suggested (Page 4.line 22)

13. page 5 line 1 - most vs mostly and also please provide rough proportions for seasonal precipitation

**Response:** Done as suggested (Page 5.line 12)

14. page 5 line 10 "Pingjiang watershed and are 2.3 to 20.1 m3/s in the Xiangshui watershed.

**Response:** Done as suggested (Page 5.line 2-3)

15.  page 5 line 21 Stream flow data are available from 1957 to 2014 for both watersheds.

**Response:** Done as suggested (Page 5.line 21)

16.  page 6 line 16 to end of page font appears smaller

**Response:** We have checked it again. They are consistent with the previous font size (Page 6.line 6 to end of page)

17. page 8 line 2 - median is not capitalized

**Response:** Done as suggested (Page 8.line 2)

18. page 10 line 13 p=0.21 and 0.27 ? How are these p-values associated with a marginally higher median flow?

**Response:** Thank the reviewer for this concern. This statement has been revised to eliminate confusion (Page10. line13).

19. page 10 section 4.3 - would benefit from details on regrowth metrics such as canopy closure or height and a comparison between basins.

**Response:** We agree that such data would be useful for improved comparison. Unfortunately, because our studied watersheds are more than 1000 $km^2$ with complex terrains and abundant vegetation types, it is very difficult to obtain long-term tree growth metrics.

20. page 12 line 2- s after enhance and reduce –

**Response:** Done as suggested (Page 12.line 2)

21. page 14 - line 2 - what was the positive role for low flows?

**Response:** Thank the reviewer for this concern. In the Xiangshui watershed, the averaged magnitude of low flow in the reforestation period was higher (p=0.084) than that in the deforestation period. Thus, we conclude that the low flows after reforestation in the Xiangshui watershed were also improved (p＜0.1).

22. page 12 line 21-South Africa

**Response:** Done as suggested (Page12. line21)

23. page 12, line 21-However, in humid regions….- as a side comment curious that increase in vegetation leads to greater net precipitation

**Response:** Indeed, hydrological responses may be inconsistent in different climatic regions especially for low flows. Thus, more case studies are needed before a general conclusion between reforestation and low flows can be developed.

24. page 13 line 7 - between the two studied....

**Response:** Done as suggested (Page13. line7)

25. page 14 line 5. Careful with this statement because watershed characteristics/properties such as slope have long been recognized as having a significant influence on runoff.

**Response:** Thank the reviewer for this suggestion. Indeed, slope has long been recognized to have a significant influence on runoff (mean flow). In this paper, we compared streamflow at different percentiles (e.g., 5%, 10%, 50% and 95%) in responses to different watersheds with different slopes.